# EgoSim:
# An Egocentric Multi-view Simulator and Real Dataset for Body-worn Cameras during Motion and Activity

**Dominik Hollidt, Paul Streli, Jiaxi Jiang, Yasaman Haghighi,**
**Changlin Qian, Xintong Liu, and Christian Holz**

Department of Computer Science
ETH Zürich, Switzerland

`firstname.lastname@inf.ethz.ch`

## Abstract

Research on egocentric tasks in computer vision has mostly focused on head-mounted cameras, such as fisheye cameras or embedded cameras inside immersive headsets. We argue that the increasing miniaturization of optical sensors will lead to the prolific integration of cameras into many more body-worn devices at various locations. This will bring fresh perspectives to established tasks in computer vision and benefit key areas such as human motion tracking, body pose estimation, or action recognition—particularly for the lower body, which is typically occluded.

In this paper, we introduce *EgoSim*, a novel simulator of body-worn cameras that generates realistic egocentric renderings from multiple perspectives across a wearer's body. A key feature of EgoSim is its use of real motion capture data to render motion artifacts, which are especially noticeable with arm- or leg-worn cameras. In addition, we introduce *MultiEgoView*, a dataset of egocentric footage from six body-worn cameras and ground-truth full-body 3D poses during several activities: 119 hours of data are derived from AMASS motion sequences in four high-fidelity virtual environments, which we augment with 5 hours of real-world motion data from 13 participants using six GoPro cameras and 3D body pose references from an Xsens motion capture suit.

We demonstrate EgoSim's effectiveness by training an end-to-end video-only 3D pose estimation network. Analyzing its domain gap, we show that our dataset and simulator substantially aid training for inference on real-world data.

EgoSim code & MultiEgoView dataset: `https://siplab.org/projects/EgoSim`

## 1 Introduction

The newest generation of AI-based personal devices evidently requires an understanding of the world from a user's perspective to provide meaningful context. For example, Meta's Ray-Ban glasses [1], Hu.ma.ne AI pin [2], or the glasses demoed at Google I/O 2024 all share the wearer's perspective to analyze their surroundings. Such emerging devices in addition to existing immersive Mixed Reality platforms have further spurred research efforts on egocentric perception tasks [3, 4].

While head-worn cameras have primarily been used for localization [5, 6], they are ideally positioned to simultaneously capture the wearer's arm motions, for example, to estimate upper body poses [7–9] or detect user input from hand poses and actions [10, 11]. For egocentric pose estimation, previous work has commonly used head-mounted fisheye cameras pointing down [12–14], which can capture much of the upper body. This promise has spurred interest in egocentric pose estimation, for which

38th Conference on Neural Information Processing Systems (NeurIPS 2024) Track on Datasets and Benchmarks.

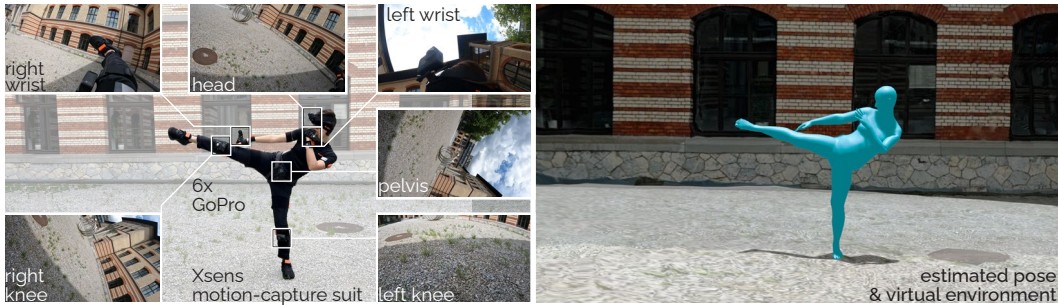

Figure 1: (Left) Our dataset *MultiEgoView* contains 5 hours of egocentric real-world footage from 6 body-worn GoPro cameras and ground-truth 3D body poses from an Xsens motion capture suit as well as 119 hours of simulated footage in high-fidelity virtual environments on the basis of real motion capture data and associated 3D body poses. (Right) Our method estimates ego poses from video data alone, here visualized inside the scanned 3D scene.

several real [3, 5, 12–18] and synthetic [13, 15, 19] datasets have been collected. The advantage of synthetic data has been demonstrated for simultaneous localization and mapping (SLAM [20–22]), 3D reconstruction [23, 24], and human mesh recovery (HMR [25–27]).

For more comprehensive capture of body motion, prior work has used motion-capture suits [28] or individual body-worn motion sensors [29–32], where learned methods predict 3D body poses from up to a set of inertial sensors as input. These sensor ensembles provide rich information about the various limb motions and enable fine-grained pose estimation. However, estimates from motion sensors alone suffer from drift and struggle with tracking global positions, for which previous work has added head-worn cameras [6, 16, 33, 34] to complement inertial motion cues.

Considering the ongoing miniaturization of camera technology, there is promise in further augmenting on-body tracking methods with camera sensors, for example, to remove the occlusion of lower body parts and extend the coverage of the environment [14]. Indeed, Shiratori et al.'s pioneering effort to track 3D body poses in the wild from multiple body-worn cameras in 2011 predates many learning-based methods [35] and demonstrated the potential of the richer modality that is videos for human motion tasks. In addition, body-worn cameras, such as those on the wrists [36–39] or legs [35] benefit from their proximity to the point of interest during human activity or hand-object interaction. The use of multiple cameras mitigates the effect of occlusion and provides multiple vantage points of the ego-body, surrounding people, and the environment. Extensive research on integrating multi-view data (e.g., [40–42]), albeit typically from static third-person perspectives, has shown benefits for navigation [43], 3D reconstruction [44], and pose estimation [45, 46].

In this paper, we introduce *EgoSim*, a multi-view body-worn camera simulator designed for human motion tasks. We also present *MultiEgoView*, a dataset that comprises rendered footage simulated from existing human motion data and novel complementary real-world recordings (Figure 1). We demonstrate the benefit of body-worn cameras and our simulator with the example of ego-body pose estimation using an end-to-end trained vision-only model. Our contributions in this dataset paper are:

1. EgoSim, an easy-to-use, adaptable, and highly realistic simulator for multiple body-worn cameras that uses real human motion as input. Camera positions on the body and their intrinsics can be configured flexibly, and EgoSim renders a range of useful modalities. EgoSim also simulates the attachment of body-worn cameras realistically via a spring arm to include motion artifacts.
2. MultiEgoView, a 119-hour video dataset of one or more avatars that perform natural motions and activities based on AMASS [47] in four virtual environments with reference 3D body poses. We contribute a novel 5-hour real-world dataset with 13 participants who wore 6 GoPro cameras with 3D body reference poses (from Xsens [48]) and dense human activity classes (BABEL [49]).
3. A learning-based multi-view method for end-to-end 3D pose estimation tasks from video. We analyze the sim2real gap based on our dataset and show the benefits of simulated data.

Taken together, we believe that EgoSim—alongside other emerging simulators (e.g., for faces [50] and scene interactions of human bodies [19, 51–54], and hands [55, 56])—will contribute to advancing open research on egocentric perception tasks.

## 2 Related Work

**Synthetic datasets and simulators.** The advancement of deep learning in recent years has necessitated larger and more varied datasets that can be acquired using simulated data. Visual synthetic data proved its benefits in many fields such as human mesh recovery [26], visual-inertial odometry [57], visual SLAM [58, 59], and human pose estimation [26, 60]. Microsoft AirSim [61] stands out as one of the most effective simulators. It has facilitated the creation of photo-realistic datasets such as TartanAir [20], optimized for Visual SLAM tasks, and Mid-air [62], designed for low-altitude drone flights. So far, AirSim [61] and other simulators [63] fall short in tasks centered on human dynamics, such as 3D human pose estimation or multi-actor interactions. Only recently, the Habitat 3 [64] simulator targets human-robot interaction tasks and progresses in this area but offers limited configuration for sensor placement and environmental diversity. EgoGen as a novel human-centered simulator demonstrates promise by focusing on human motion synthesis [19]. Traditionally, datasets simulate cameras either statically or with smooth movements. Such datasets fail to generalize to egocentric scenarios where the camera's position dynamically changes in relation to the wearer's movements. EgoSim advances this field by being specifically designed for human-centric research with wearable cameras that follow the natural non-smooth movements within the human body. It uniquely supports complex multi-character interactions in varied environments, both indoor and outdoor, enabling more comprehensive and diverse studies in this field.

**Human motion datasets.** In controlled settings, multiple third-person view cameras and motion capture equipment offer accurate ground-truth data [47, 65–69]. Fitting 3D body models [70–72] to point cloud marker sets [47] or using RGBD camera data can provide ground-truth poses. However, the complexity of these setups mostly limits their scalability to indoor environments [3, 73, 74]. Pseudo-ground truth pose annotations can overcome these limitations for outdoor environments. Several methods use 2D keypoints [75–77], which are easy to label at a large scale, but provide 2D constraints only on the human pose. Alternatively, fitting 3D body models such as SMPL [70] to images provides pseudo-ground truth parameters [78–80]. You2Me [81] and EgoBody [3] capture human pose data for interacting individuals using head-mounted cameras in indoor settings. Recently, Egohumans [4] has expanded the scope to include up to four interacting individuals in both indoor and outdoor settings. Meanwhile, larger datasets like Ego4D [17, 18] offer extensive data from head-mounted cameras for tasks such as social interaction and hand-object interaction, but they lack data from additional body-worn cameras. The recently published Nymeria dataset [82] addresses this gap partially and includes real-world videos from wrist-mounted cameras. Our real-world MultiEgoView dataset further extends to a setup with six body-worn cameras with additional sensors at the knees and pelvis. To overcome limitations in real-world datasets, realistic synthetic datasets offer an alternative that offers diversity and quality ground truth annotations [15, 26, 27]. Our work expands on this approach by introducing a configurable simulator tailored to body-worn sensors, with adjustable parameters for lighting, scene, and camera placement. EgoSim complements real-world datasets like Nymeria by enabling the rendering of synthetic images from adjustable body-worn cameras based on their captured motion sequences.

**Egocentric perception.** Wearable cameras serve as the primary input for research on egocentric perception tasks. Currently, real and synthetic egocentric datasets mainly feature head-mounted sensors. Some systems [13, 14, 83] use a single head-mounted, body-facing, fisheye camera to estimate 3D ego-body pose, while others rely on a stereo configuration [15, 73, 74]. Head-mounted, body-facing cameras benefit from capturing visible joints in image space to aid ego-body pose estimation. Other methods recover the 3D pose from non-body-facing cameras. HPS [84] integrates multiple body-worn IMUs with camera-based localization using structure from motion. Kinpoly [85] recovers the whole body pose from a front-facing camera using physics simulation with reinforcement learning, while EgoEgo [5] combines SLAM with a diffusion model to recover the ego-body pose. AvatarPoser [33] and its subsequent work [8, 34] predict full-body poses based on head and hand poses tracked by commercial mixed reality devices. HOOV [86] extends hand tracking beyond the field of view of head-mounted cameras using inertial signals captured at the wrist.

So far, egocentric datasets have mainly focused on head-mounted cameras that either point down toward the body [12, 14, 15] or forward [5, 16, 87], often designed for specific devices [3, 4, 18, 88]. Our work extends egocentric datasets to multiple body-worn cameras by providing an adaptable simulation platform and a real-world dataset of six body-worn cameras.

Table 1: Comparison of previous datasets for egocentric 3D human pose estimation.

| Dataset | Mo2Cap2 [12] | xR-EgoPose [13] | EgoCap [73] | EgoGlass [74] | UnrealEgo [15] | EgoBody [3] | ARES [5] | MultiEgoView (ours) |
|---|---|---|---|---|---|---|---|---|
| Head camera type | fisheye | fisheye | wide stereo | stereo | fisheye stereo | front facing | fisheye stereo | front facing |
| sees body | ✓ | ✓ | ✓ | ✓ | ✓ | ✗ | ✗ | partly |
| Hand camera | ✗ | ✗ | ✗ | ✗ | ✗ | ✗ | ✗ | 2× |
| Leg camera | ✗ | ✗ | ✗ | ✗ | ✗ | ✗ | ✗ | 2× |
| Pelvis camera | ✗ | ✗ | ✗ | ✗ | ✗ | ✗ | ✗ | ✓ |
| Image generation | composite | Maya | composite | real | Unreal Engine | real | Replica | Unreal Engine |
| Image quality | low | high | real | green screen | high | real | high | high |
| Environment | In- & Outdoor | Mostly Indoor | Indoor | Indoor | In- & Outdoor | Indoor | Indoor | Outdoor |
| Dataset Size | 530$k$ | 383$k$ | 2 × 41k | 2 × 170k | 2 × 450k | 220$k$ | 1.2$M$ | 6 × 12.9$M$ |
| Real data | ✗ | ✗ | ✓ | ✓ | ✗ | ✓ | ✗ | ✓6 × 520$k$ |
| Motion Diversity | mid | low | low | low | high | high | high | high |

## 3   EgoSim Simulation Platform

EgoSim is designed for body-worn camera simulation. We extend Microsoft's AirSim simulator [61] integrated within the Unreal Engine [89] to leverage its flexibility and realistic output renders (e.g., [26, 27]). Specifically, we augment the platform with the capability of simulating *body-worn* cameras during realistic human motion, generating dynamic changes in camera motion that correspond to a person's movements, including potentially irregular, rough, and non-smooth moments.

**Simulating images.**   EgoSim renders footage through Unreal Engine's cinematic camera [90] for realistic images. The camera model and noise parameters are adjustable. EgoSim supports simultaneously rendering multiple modalities (Figure 2), including RGB, depth, normal maps, and semantic segmentation masks. These modalities are complementary and can serve as input to various computer vision tasks in the future.

**Simulating physical attachment and motion artifacts.**   A key feature of EgoSim is the consideration of camera attachment to account for motion artifacts during simulation. Since body-worn cameras are non-rigidly mounted, often coupled to clothing or strapped to the limbs like a smartwatch, the loose attachments can lead to slip and drag in the camera's position and orientation. EgoSim simulates these using spring arm mounts that connect the avatar's body and the virtual cameras. We demonstrate that spring-damper systems as a camera mounting model help to realistically capture the effects of loose camera attachment as found in the real world (Section 6.3).

**Simulating diverse environments.**   EgoSim benefits from the vast selection of indoor and outdoor environments available in Unreal and previous work, e.g., [26]. As shown in Figure 3, it can render both, large, realistic hand-modeled scenes and scanned scenes that closely resemble their real-world counterparts. The used scenes are in wide open spaces where motion capture is traditionally hard to perform. Additional details about EgoSim's features are provided in the appendix Table 3.

**Synchronizing multi-sensor and multi-person recordings.**   Synchronizing multiple cameras poses challenges in real-world recordings, yet it is straightforward to generate synchronized multi-modal data in EgoSim while obtaining ground-truth characteristics of the environment or avatars. In addition EgoSim is capable of simulating and rendering data from *across* multiple avatars and to obtain corresponding ground-truth poses and camera positions. EgoSim supports a flexible number of sensors, sensor characteristics, and attachment locations—independently for each avatar.

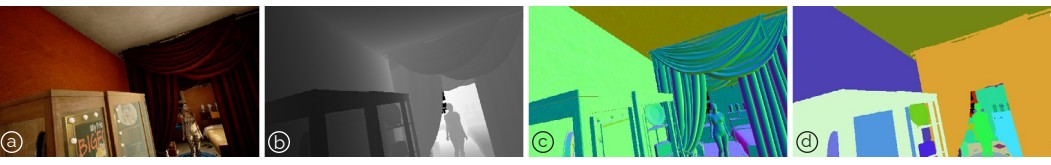

Figure 2: EgoSim renders multiple modalities: (a) RGB, (b) depth, (c) normals, (d) semantic labels.

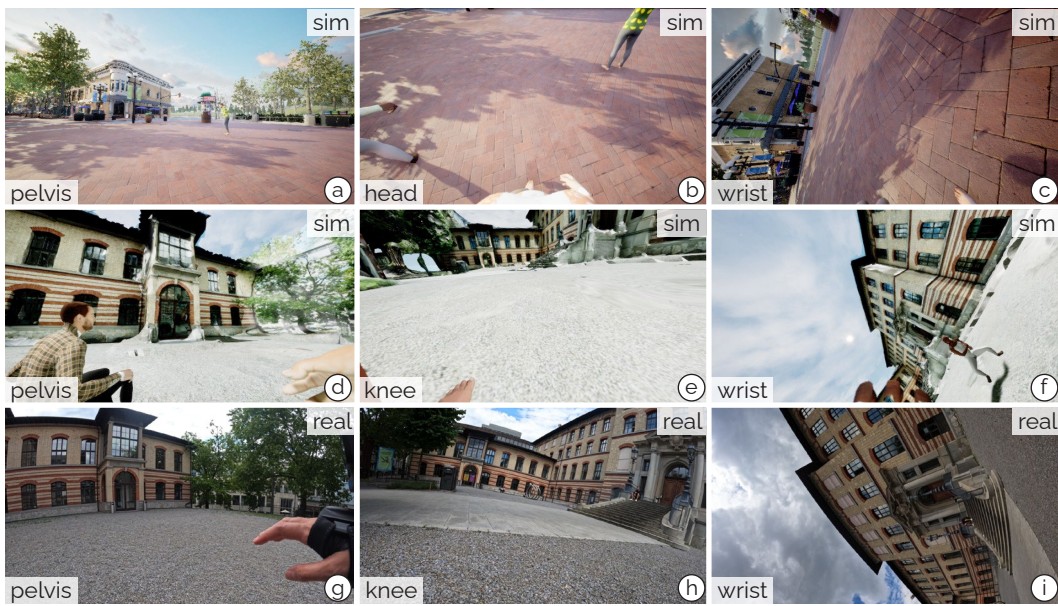

Figure 3: Example RGB renders produced by EgoSim and included in our MultiEgoView dataset. Qualitatively, the simulated scan (d, e, f) and real data (g, h, i) look similar. Both simulated scenes (Scene 1: a, b, c; Scene 2: d, e, f) offer high-fidelity environments. The pelvis provides a stable view of the environment, whereas wrist and knee cameras typically move quickly and capture artifacts.

## 4 MultiEgoView Dataset

MultiEgoView contributes a sizeable and synchronized dataset of RGB data from six body-worn cameras, along with ground-truth body poses and activity annotations. Our dataset includes real and synthetic data, providing a challenging and interesting testbed for training and benchmarking body-pose estimation, activity classification, dynamic camera localization, and mapping algorithms.

**Synthetic data generation.** Using EgoSim, we rendered a dataset of 77.4 M RGB images corresponding to 119.4 hours captured by six virtual cameras on a virtual avatar. Images were rendered with a 118° field of view (FOV) at a resolution of $640 \times 360$ and a framerate of 30 fps. Cameras were attached to the head, pelvis, wrists, and knees, facing outwards to capture both the environment and parts of the wearer's body. This considerably extends the focus of prior work on head-mounted cameras [12, 13, 15, 73] and better resembles emerging wearable platforms devices [3, 17, 74]. To ensure realistic motions, we animated avatars using motion capture sequences from AMASS [47], converted to FBX format for EgoSim support [71]. We randomly varied avatar appearances in terms of skin color and clothing texture using BEDLAM's assets [26]. Our dataset features 24 locations across 4 scenes: (1) a hand-built virtual outdoor environment of a city, (2) the front courtyard of a university building that we scanned using Polycam, with an accurate public point cloud scan and structure-from-motion model available [91], (3) Downtown city with skyscrapers and (4) a park with sport courts, lawn, vegetation and water. Each scene includes up to four simultaneously animated avatars to increase diversity and support multi-view multi-human pose estimation [4, 92]. In addition to the RGB data, we provide ground-truth camera and 3D avatar poses, as well as simulated accelerometer and gyroscope readings from all six cameras.

**Real-world data collection.** We captured a dataset of ∼5 hours in the real world using six GoPro cameras (5×HERO 10, 1×HERO 9) [93], worn at the same body locations as in our simulation. We recruited 13 participants from places around our institution for this collection, who consented to participation and data recording. The study considers ETH ethics guidelines and Participants received a small gratuity for their time. Data was recorded in the same university front courtyard that was scanned for the synthetic environment (2), using GoPros set to a resolution of 1080p at 30 fps and a horizontal FOV of 118°. The 13 participants (4 female, 9 male, ages 21–30, mean = 26.4) were recruited from our institution, with heights ranging from 160–190 cm (mean = 176.1, SD = 9.5) and

weights from 50–94 kg (mean = 69.6, SD = 13.2). After providing consent (see Appendix for details), participants were equipped with a full-body Xsens [48] motion capture suit for ground-truth pose capture. Following an initial calibration, participants performed a block of 35 different activities featuring the most common motions from AMASS according to the BABEL annotation [49]. For a full list of activities, see our appendix Table 7. Each block lasted about 10 minutes, with participants repeating the block 1–3 times. To synchronize the GoPro camera with Xsens, participants clap at the beginning of each recording and match the camera with extracted SMPL poses. We compute shape parameters from the body measurement of participants with Virtual Caliper [94].

All sequences across real and synthetic data are labeled with activity classes from BABEL [49].

## 5 Baseline Method: Wearable Multi-Camera Body Pose Estimation

To demonstrate the benefits of MultiEgoView, we trained a neural network to estimate 3D ego body poses using multiple body-worn cameras. The input to the network consists of the aligned video sequences $\boldsymbol{X} \in [0, 1]^{C \times F \times 3 \times H \times W}$, with $F$ frames from $C$ body-attached cameras. Based on these inputs, the network predicts a pose $\hat{\boldsymbol{p}}_i$ for each input frame $i$.

### 5.1 Network architecture

Our network is a Vision Transformer Model based on Sparse Video Tube ViTs [95]. We extract feature vectors from each input video using a sparse view tokenizer SVT with a shared interpolated kernel. The extracted feature vectors from the sparse tube tokenizers are then added to their fixed spatio-temporal position encoding $\boldsymbol{\kappa}_p$ and their learnable view encoding $\boldsymbol{\kappa}_{v,c}$ per camera $c$.

$$\boldsymbol{v}_c = \text{SVT}(\boldsymbol{X}_c, \boldsymbol{W}) + \boldsymbol{\kappa}_p + \boldsymbol{\kappa}_{v,c}, \quad \text{where } \boldsymbol{W} \text{ are the shared weights of the kernel.} \tag{1}$$

The resulting feature vectors for the different cameras $\boldsymbol{v}^c$ are concatenated with the pose token $\boldsymbol{\phi}_j = \boldsymbol{\tau}(j) + \boldsymbol{\psi}, j \in [0, F-1]$, where $\boldsymbol{\psi}$ is a trainable pose token and $\boldsymbol{\tau}$ is a sinusoidal positional encoding. The resulting token sequence is then processed using a Vision Transformer Encoder.

$$\{\boldsymbol{z}_0, \ldots, \boldsymbol{z}_{F-1}\} = \text{ViT}(\text{concat}(\boldsymbol{\phi}_0, ..., \boldsymbol{\phi}_{F-1}, \boldsymbol{v}_0, ..., \boldsymbol{v}_{c-1})) \tag{2}$$

Based on each embedded pose token $\boldsymbol{z}$, we obtain the 6D representation [96] of the SMPL pose parameters $\boldsymbol{\theta}$, the 6D relative rotation $\boldsymbol{R}_r$, and 3D relative translation of the root $tr$ with respect the previous frame.

$$\hat{\boldsymbol{\theta}} = W_\theta \boldsymbol{z}, \quad \hat{\boldsymbol{R}}_r = W_R \boldsymbol{z}, \quad \hat{\boldsymbol{t}}_r = W_t \boldsymbol{z} \tag{3}$$

To improve generalization, the network is trained to predict the pose difference, i.e., the relative root pose with respect to the previous pose, instead of directly predicting global root poses.

Using Forward Kinematics, we obtain the global body pose $\boldsymbol{p}$ with respect to the starting pose.

$$\{\hat{\boldsymbol{p}}_0, \ldots, \hat{\boldsymbol{p}}_{F-1}\} = \text{FK}_\theta(\boldsymbol{\theta}, \hat{\boldsymbol{R}}_g, \hat{\boldsymbol{t}}_g, \beta), \text{where} \quad \hat{\boldsymbol{R}}_g, \hat{\boldsymbol{t}}_g = \text{FK}_g(\hat{\boldsymbol{R}}_r, \hat{\boldsymbol{t}}_r) \tag{4}$$

Where $\beta$ are the shape parameters of the SMPL-X model [71] for a given person.

We use 4 tubes with the following configurations: $16 \times 16 \times 16$ with stride $(12, 48, 48)$ and offset $(0, 0, 0)$, $24 \times 6 \times 6$ with stride $(12, 32, 32)$ and offset $(8, 12, 12)$, $12 \times 24 \times 24$ with stride $(24, 48, 48)$ and offset $(0, 28, 28)$, and $1 \times 32 \times 32$ with stride $(12, 64, 64)$ and offset $(0, 0, 0)$. The pose embedding parameter is initialized using the Kaiming uniform distribution [97], and the pose token is initialized using the Normal distribution.

### 5.2 Loss function

We supervise the network with the following loss function:

$$\mathcal{L} = \lambda_\theta \mathcal{L}_\theta + \lambda_p \mathcal{L}_p + \lambda_v \mathcal{L}_v + \lambda_{t_r} \mathcal{L}_{t_r} + \lambda_{R_r} \mathcal{L}_{R_r} + \lambda_{t_g} \mathcal{L}_{t_g} + \lambda_{R_g} \mathcal{L}_{R_g} + \lambda_z \mathcal{L}_z \tag{5}$$

The angle loss $\mathcal{L}_\theta$ encourages the model to learn the SMPL angles $\boldsymbol{\theta}$, while the joint position loss $\mathcal{L}_p$ forces the predicted joint positions through forward kinematics to be close to the ground-truth joint positions. This way, both the local and the accumulated errors are considered.

$$\mathcal{L}_\theta = |\boldsymbol{\theta}_{6\text{D}} - \hat{\boldsymbol{\theta}}_{6\text{D}}|_1 \quad \text{and} \quad \mathcal{L}_p = |\boldsymbol{p} - \hat{\boldsymbol{p}}|_1, \tag{6}$$

where *6D* indicates the six-dimensional representation of the rotation matrices [96]. For the root pose, we penalize both the relative and absolute translation and orientation error accumulated through the kinematic chain,

$$\mathcal{L}_{R_r} = |\mathbf{R}_{r,6\mathrm{D}} - \hat{\mathbf{R}}_{r,6\mathrm{D}}|_1 \quad \text{and} \quad \mathcal{L}_{t_r} = |\mathbf{t}_r - \hat{\mathbf{t}}_r|_1$$
$$\mathcal{L}_{R_g} = \|\hat{\mathbf{R}}_g \mathbf{R}_g^{-1} - I\|_2 \quad \text{and} \quad \mathcal{L}_{t_g} = |\mathbf{t}_g - \hat{\mathbf{t}}_g|_1 \tag{7}$$

To encourage the model to estimate more expressive motions accurately, we add a velocity loss $\mathcal{L}_v$. We also regularize the embedded pose token $\mathbf{z}$ using an $l_2$-regularization term $\mathcal{L}_z$.

$$\mathcal{L}_v = |(\mathbf{p}_i - \mathbf{p}_{i-1}) - (\hat{\mathbf{p}}_i - \hat{\mathbf{p}}_{i-1})|_1 \quad \text{and} \quad \mathcal{L}_z = \|\mathbf{z}\|_2 \tag{8}$$

We set $\lambda_\theta = 10$, $\lambda_p = 25$, $\lambda_v = 40$, $\lambda_{t_r} = 25$, $\lambda_{R_r} = 15$, $\lambda_{t_g} = 1$, $\lambda_{R_g} = 0.025$, and $\lambda_z = 0.0005$.

## 6 Experiments

We empirically study the effectiveness of MultiEgoView for egocentric body pose estimation. Following the BABEL-60 split [49] (60%/20%/20%), sequences of synthetic data are divided into segments of up to 5 seconds. The baseline model directly takes inputs from all six cameras, normalizes the images to the ImageNet mean, and downsamples them to $224 \times 224$ pixels at 10 fps. We accelerate the training process with a pre-trained sparse tube tokenizer on UCF101 [98,99]. The model is trained using the Adam optimizer with a learning rate of $1 \times 10^{-4}$ on an Nvidia GeForce RTX 4090 with a batch size of 12 for 135k steps, taking around 3 days.

For real data, we use a random 80%/20% split with the same 5-second chunking and training parameters. We also conduct a cross-participant evaluation, using 10 participants for training and 3 for testing, to demonstrate the model's generalization ability.

### 6.1 Quantitative metrics

We evaluate our model on a series of metrics using the body joints of the SMPLX model as follows:

**Global MPJPE** ($m$) Evaluates the mean $l_2$-norm between predicted and ground truth joint positions, punishing both pose and global position errors.

**PA-MPJPE** ($m$) Assesses pose estimation accuracy after aligning joint positions up to a similarity transform, isolating pure pose errors.

**MTE** ($m$) Mean Translation Error measures the mean $l_2$-norm of global root translation errors, indicating global translation accuracy.

**MRE** Mean Rotation Error reports global orientation error using $\|R\hat{R}^{-1} - I\|$.

**MJAE** (°) Mean Joint Angle Error compares predicted joint angle errors in degrees without considering forward kinematic chain errors.

**Jerk** ($m/s^3$) Measures the smoothness of the predicted movement, indicating temporal continuity and naturalness of motion.

### 6.2 Evaluation results

Table 2 shows the results of our multi-view pose transformer when trained on MultiEgoView. Training on synthetic data shows a low PA-MPJPE, implying a very good pose estimation. The slightly higher global MPJPE error arises due to a worse estimation of the root translation and rotation. The combination of synthetic and real data in MultiEgoView is crucial, as direct sim-2-real and training solely on real data fails to achieve accurate pose estimation. Pretraining on synthetic data followed by fine-tuning on real data improves the global MPJPE by 3.1-4 times and also lowers the PA-MPJPE by at least 2.7 cm, indicating a knowledge transfer of pose understanding from the large synthetic dataset to the real-world data. Even with a reduced fine-tuning train split of 20%, the network predicts accurate poses, though with a 8.8% increase in translation error. This showcases the benefit of synthetic data in improving pose estimation on scarce real training data. The results of the cross-participant evaluation lag behind the others. Indicating that more diversity could be required to obtain stable cross-participant results.

Table 2: Results of our method on MultiEgoView, showing the benefit of our simulator.

| Method trained on | evaluated on | Global MPJPE ↓ | PA-MPJPE ↓ | MTE ↓ | MRE ↓ | MJAE ↓ | Jerk |
|---|---|---|---|---|---|---|---|
| Synthetic | Synthetic | 0.16 | 0.040 | 0.13 | 0.272 | 9.1 | 21.9 |
| Synthetic | Real | 0.77 | 0.119 | 0.71 | 0.947 | 29.0 | 20.9 |
| Real | Real | 1.23 | 0.087 | 0.79 | 1.030 | 16.4 | 1.5 |
| *with fine-tuning:* | | | | | | | |
| Synthetic + 20% Real | Real | 0.40 | 0.056 | 0.37 | 0.504 | 12.8 | 15.4 |
| Synthetic + 80% real | Real | 0.33 | 0.044 | 0.31 | 0.415 | 10.2 | 16.7 |
| Synthetic + 10 real participants | Real (cross-participant) | 0.35 | 0.060 | 0.32 | 0.557 | 16.6 | 18.3 |

The visualization of the predicted poses in Figure 4 confirms the quantitative metrics. Generally, the model estimates the pose accurately. The biggest errors typically occur in the fast-moving limbs, as seen in the right column of Figure 4 where the model does mistakenly detect an arm movement. The pose outputs of the model are spatial-temporally smooth, which is also reflected in low jerk values (Table 2). Generally, the evaluations yielding lower Jerk indicate less active predictions that do not fully capture the full range and speed of the gt-motions (gt-jerk on eval set is 29.3) but still look natural. The model's weak point is the higher error global root position estimates. We attribute this weakness in global transformation prediction to two factors: 1) The model predicts the relative transformation between each frame, simplifying training by focusing on neighboring frames' transformations. However, small errors in relative prediction quickly accumulate in the forward kinematics. 2) The model lacks a method-based grounding of global position (e.g., through SLAM or SfM), making its transformation prediction reliant on learned environmental understanding.

Overall, MultiEgoView, with its synthetic and real-world data, shows its utility by enabling sim-to-real transfer learning. Our results also show that this would not be possible with just synthetic data or the amount of real data captured, validating the benefit of our simulator.

### 6.3 Spring Damper

Body-worn cameras experience motion artifacts, especially when mounted on limbs that move quickly, due to non-rigid mounting points. EgoSim models these motion artifacts via a Spring Arm. We demonstrate that a spring-damper system approximates real camera motion better than a rigid mount. For that, we used an OptiTrack motion capture system to track both the attached body and the GoPro that we loosely attached to the body. Using an OptiTrack motion capture system, we tracked both the body and a loosely attached GoPro. Results show the spring-damper model yields a lower mean position error (1.98 cm) compared to the rigid model (2.35 cm), highlighting its effectiveness.

## 7 Discussion

While egocentric pose estimation has been well explored, in prior implementations head-mounted cameras faces challenges such as self-occlusion, reduced resolution for lower body reconstruction,

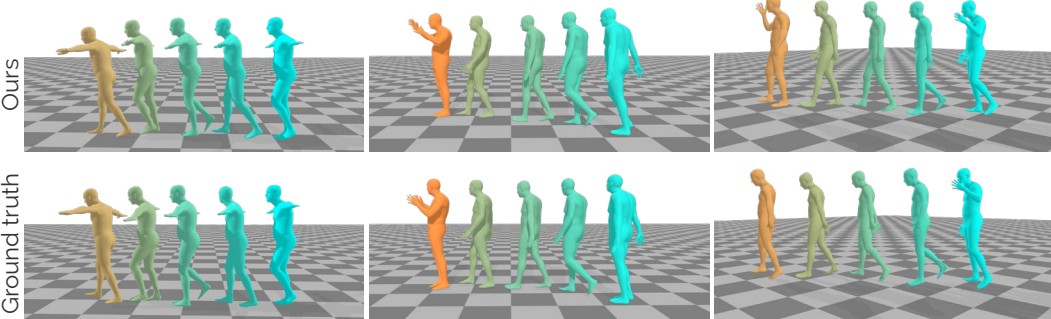

Figure 4: Visualization of our results obtained from our multi-view egocentric pose estimator on real-world data. The change of color denotes different timestamps.

and lack of environmental information. Here, multiple body-worn cameras can mitigate these by providing dynamic, multi-view perspectives that simultaneously capture the environment and the body and, more importantly, the interaction between our hands and legs with the surroundings.

EgoSim, together with MultiEgoView, is a first stepping stone to deepen our understanding of human activity from body-worn cameras at various locations. We showcase the usefulness of MultiEgoView for ego pose estimation with our learned video-based end-to-end multi-view model. Our findings show that ego pose can effectively be estimated from several body-mounted cameras and EgoSim's rendered data helps obtain better pose estimation in sim-to-real scenarios.

**Limitations of EgoSim.** Our current simulator has some limitations that will be addressed in future iterations. First, although our data includes multi-human scenarios, individual avatar animations are sampled independently from AMASS. These animations, while physically plausible, do not account for interactions with other humans or objects, limiting the study of such interactions. Additionally, our system currently features only four scenes, which can be extended to improve the generalization. Lastly, while our simulator supports high-fidelity rendering, improvements in graphics and neural rendering methods [100] are expected to reduce the simulation-to-real gap further.

**Future research on MultiEgoView.** While the pose estimation capabilities of our multi-view transformer trained using EgoMultiView are convincing on real-world data (PA-MPJPE < 5 cm), there is still room for improvement in the global position and orientation estimation of the root, especially for long sequences, where cumulative errors in root position become more pronounced. Future research directions could consider integrating low-drift camera localization methods, such as SLAM [59], or image-based localization via structure from motion [101], to achieve more stable global translation and orientation. Moreover, our current experiments only utilize RGB data. Future research could leverage MultiEgoView to enhance inertial-based pose estimation [29], depth estimation using monocular or multiple cameras [102], and semantic scene classification [103], all of which are supported by the ground-truth annotations provided by our simulator.

## 8   Conclusion

We have proposed EgoSim, an egocentric multi-view simulator for body-worn cameras that generates multiple data modalities to support emerging wearable motion capture and method development. Using EgoSim, we partially generated MultiEgoView, the first dataset that complements existing head-focused egocentric datasets with synchronized footage from six cameras worn at other locations on the body, simulated from accurate and real human motion and artifacts. We complement MultiEgoView's 119 hours of synthetic data with 5 hours of actual recordings from 6 body-worn GoPro cameras and 13 participants during a wide range of motions and activities in the wild with annotated 3D body poses and classification labels to bridge the gap between simulation and real-world data.

In the wake of the emerging area of vision-based method development from one or more body-worn sensors, we believe that our release of EgoSim and MultiEgoView will be a useful resource for future work to increase our understanding of human activities and interactions in the real world.

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

# A    Data Access

The MultiEgoView dataset, its structural description, and usage information can be found here: `https://siplab.org/projects/EgoSim`. We will release EgoSim's code to facilitate future research and data generation. An overview of EgoSim's rich customization options can be found in Table 3. An overview of the diversity of our scenes is shown in Figure 5.

Figure 5: An excerpt of our example images from 24 locations across 4 scenes.

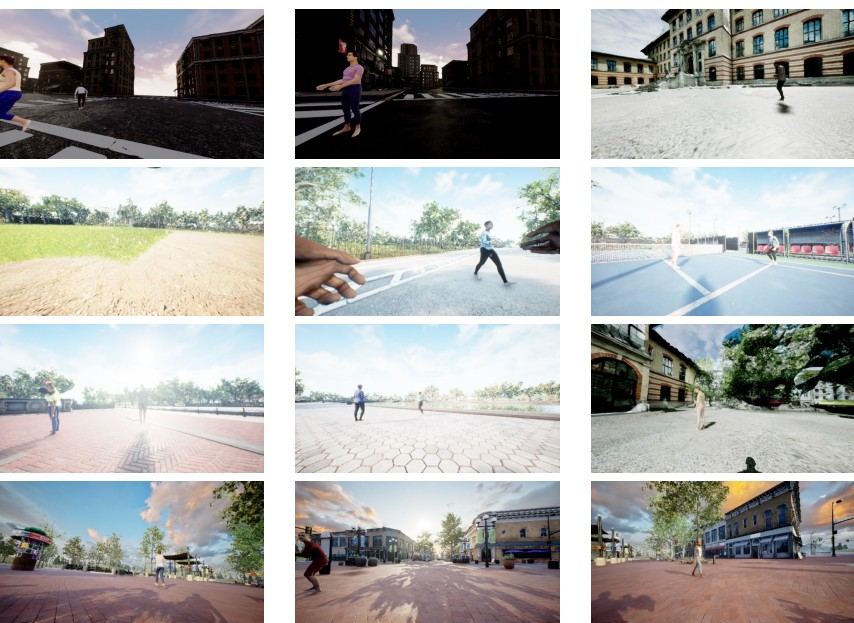

# B    Model Complexity and Ablation Studies

Our multiview transformer is ViT-based and has 114M trainable parameters and requires roughly 1.7GB VRAM for inference. With an input/output window of up to 5 seconds, the inference time is 17.6ms on an RTX 4090 with a batch size of 1, making the system real time capable. Training with 6 cameras and a batch size of 12 increases VRAM demands to 20GB.

## B.1    Analysis of Cameras

Previous work utilized varying numbers of body-worn cameras [5, 6, 35, 85]. In our ablation study (Table 4), we demonstrate the benefits of using more cameras. In this ablation study, we use scenes (1) and (2) of our dataset. Our multiview transformer achieves the lowest global-MPJPE with six cameras. Even with fast-moving cameras (see Section C) attached to knees and writs, our method accurately recovers body pose, though global translation error increases. Using only head and wrist cameras results in higher pose errors, particularly in leg and foot movements ($0.068/0.106/0.145m$ root-aligned foot position error for the three configurations respectively). The use of additional cameras also leads to more pronounced and active motions. The model tends to average poses over sequences rather than capturing rapid movements. This is evidenced by a decreased jerk with fewer cameras, a finding further supported by qualitative analysis.

This highlights the advantage of additional cameras, especially for accurately estimating limb poses, even when attached to fast-moving mounting points. Thus, showing that we do not require cameras on stable positions, e.g. head or pelvis.

Table 3: Details of EgoSim's features that allow simulating complex scenarios for body-worn sensors in egocentric settings. These features are especially useful in contexts where data is scarce and data collection poses significant challenges or requires extensive time.

| EgoSim Feature | Description |
| --- | --- |
| Avatar skeletal mesh options | Compatible with SMPL [70], SMPL+H [104], SMPL-X [71], and custom skeletal meshes via the FBX format |
| Avatar motion capabilities | Supports MoCap data [26, 47] as well as synthetic motions |
| Camera customization | Adjustable image resolution, field of view (FOV), and auto exposure settings including speed, bias, brightness limits, and spring system between body and camera for realistic motion simulation of nonrigid camera mounting |
| Image noise and distortion | Customizable noise intensity and horizontal bump distortion |
| Environmental settings | Support for various environments including indoor and outdoor settings, diverse weather conditions, and lighting variations based on Unreal Engine [89] marketplace |
| Egocentric and external camera integration | Support for both egocentric cameras attachable to different body parts and stationary external cameras to facilitate third-person perspective captures. |

Table 4: Results of different camera setups. Adding more cameras yields better pose prediction. Training and evaluation were conducted on synthetic scenes (1) and (2).

| cameras | Global MPJPE | PA-MPJPE | MTE | MRE | MJAE | Jerk |
| --- | --- | --- | --- | --- | --- | --- |
| all six | 0.18 | 0.041 | 0.14 | 0.334 | 9.3 | 21.7 |
| wrists & knees | 0.238 | 0.051 | 0.197 | 0.376 | 11.1 | 19.4 |
| head & wrists | 0.293 | 0.06 | 0.243 | 0.454 | 11.6 | 14.6 |
| head | 0.345 | 0.0823 | 0.286 | 0.452 | 14.7 | 0.9 |

## B.2 Analysis of Scenes

Within the main paper, we investigated the sim-to-real transfer of our model. Here, we investigate the model's ability to transfer its knowledge between scenes. Table 5 shows that there is a significant rise in pose prediction error when transferring scenes. Interestingly the model is generally able to predict the root pose (low MTE) and lower body pose of the avatar while the arms are badly predicted, as confirmed by a qualitative inspection. The trend is similar when training on just 1 scene and evaluating on the scene (2) and when training on (1) and (2) and evaluating on the very diverse scenes (3) and (4).

This indicates the opportunity for future scene generation to improve the dataset's generalizability. As we will publish EgoSim upon acceptance, future research can tailor the synthetic scenes to achieve maximal performance in the target domain.

## C Analysis of Camera Positions

Limb-based cameras, such as those mounted on wrists or knees, experience higher velocities, accelerations, and jerks, making them harder to track and localize. As shown in Table 6, the head and pelvis are the most stable mounting points, with the least movement. In contrast, wrists have the highest average acceleration due to rapid arm movements during activities like walking. Knees follow slightly behind as they are mostly steady for all standing motions. Overall, MultiEgoView offers many body camera positions with varying stability.

Table 5: Results of scene transfer.

| Train Scene | Eval Scene | Global MPJPE | PA-MPJPE | MTE | MRE |
|---|---|---|---|---|---|
| Downtown (1) | Downtown (1) | 0.18 | 0.043 | 0.15 | 0.270 |
| Downtown (1) | CAB (2) | 0.37 | 0.143 | 0.28 | 0.466 |
| Synthetic (1) & (2) | Synthetic (1) & (2) | 0.18 | 0.041 | 0.14 | 0.334 |
| Synthetic (1) & (2) | Synthetic (3) & (4) | 0.42 | 0.148 | 0.34 | 0.47 |

Table 6: Statistics about the movements of the real-world data of MultiEgoView. The head and pelvis offer the most stable positions on the body, while wrists and knees experience much higher average accelerations and changes of acceleration.

| Joints | mean velocity $(m/s)$ | mean acceleration $(m/s^2)$ | mean jerk $(m/s^3)$ |
|---|---|---|---|
| Head | $0.53 \pm 0.12$ | $2.37 \pm 1.80$ | $113.46 \pm 192.24$ |
| Pelvis | $0.48 \pm 0.12$ | $2.39 \pm 1.77$ | $119.46 \pm 192.24$ |
| Wrists | $0.83 \pm 0.15$ | $4.95 \pm 2.28$ | $163.4 \pm 241.90$ |
| Knees | $0.61 \pm 0.12$ | $3.58 \pm 1.69$ | $162.4 \pm 187.18$ |

## D   Data Recording Procedure

Participation in the data recording was entirely voluntary. Participants were required to sign a consent form for both the data recording and the subsequent publication of the data. They retained the right to withdraw their consent for recording and publication at any time before or during the data collection process. The names and identities of the participants will remain confidential and undisclosed. As a token of appreciation, participants received a small gift for their involvement in the study.

Upon obtaining informed consent, participants were given a brief overview of the recording procedure and the specific movements required for the study. The recording session commenced from a standardized starting position, followed by a brief calibration process. Cameras were then activated and synchronized by a clap. Participants performed the prescribed movements within a predefined area, executing them in a sequential order. To introduce variability in both position and camera perspectives, participants were instructed to take one to five steps between each movement repetition. The recording session concluded with participants returning to the initial starting position. A recording session took on average 10:28 minutes with a standard deviation of 1:50 minutes, up to 3 sessions were recorded per participant.

## E   Data Annotation

To gain insights into the semantics of human movement, we manually annotated the real-world recordings following the categories from BABEL [49]. BABEL densely annotated the majority of the AMASS dataset with action labels. Annotators identified segments and assigned labels to these segments. The raw, language-based annotations were then categorized into action categories. Building on the BABEL framework, we included commonly found movements from BABEL in our recordings, see Table 7.

Our annotations cover the entire sequence from the starting position to the return to the starting position. During the recording, participants performed different movements sequentially, often walking a few steps between motions. These intermediary steps were not annotated as separate segments unless they exceeded a few steps.

Most action classes are featured for around 6 minutes in the dataset. Walking is the most prominent as participants often walk between different movements. MultiEgoView features a wide coverage of different movements from leg and arm motions to sports activities, making it an ideal resource for evaluating BABEL-based systems on real-life data.

Table 7: Overview of different movements in the real-world data of MultiEgoView. MultiEgoView covers a wide range of 35 different movements.

| Motion | Fraction (%) | Motion | Fraction (%) |
|---|---|---|---|
| jump | 3.18 | stretch body left right | 2.56 |
| walk | 11.68 | wave arm | 2.48 |
| A-pose | 2.95 | bicep curls | 2.51 |
| kick ball | 2.88 | elbow to opposite knee | 2.44 |
| throw ball | 3.07 | raise left/right arm | 2.46 |
| stand | 3.80 | arms in front of chest | 2.50 |
| dribble ball | 2.58 | squats | 2.35 |
| side steps | 3.41 | T-pose | 3.75 |
| aim with hand | 2.84 | arms over head | 2.34 |
| rotate arms | 3.0 | walk backward | 2.38 |
| move arms to front | 2.38 | balance step feet in one line | 2.60 |
| lunge with arms to the side | 3.11 | pick something up one arm | 2.69 |
| lunge | 2.26 | pick something up both arms | 1.81 |
| punch the air in front | 2.49 | blow kiss | 1.91 |
| walk with extended arms | 2.53 | bow | 1.83 |
| swing tennis racket | 2.77 | crouch down | 2.04 |
| arms to face | 2.34 | jumping jacks | 1.63 |
| stretch arms left and right | 2.27 | | |

# F   Ethical Considerations

EgoSim's high-fidelity simulation of camera footage addresses several ethical implications associated with motion capture, particularly in real-world settings. Motion capture is afflicted by privacy concerns for recorded individuals, especially given the need for larger-scale capture of representative human data with diverse participants. Our simulator mitigates this by synthesizing data from realistic avatars whose appearances can be flexibly adjusted while expressing behavior based on actual human motion. This not only preserves individual privacy but also allows the creation of diverse datasets that include a wide range of ethnic backgrounds, which is crucial for the effective generalization of learned algorithms. Consequently, our simulator provides a valuable tool for advancing real-world perception inference while respecting ethical considerations.

Body-worn cameras capture extensive environmental details, offering the potential for simultaneous ego-body and environment understanding. Capturing data with more cameras always opens up more opportunities for surveillance. In our case, the amount of cameras results in the unintended exposure of individuals in the proximity of the participant to data recording. MultiEgoView's focus is on the ego-body, therefore we minimize the exposure of other people in the dataset by selecting a recording area with a limited number of passersby. Additionally, to protect the privacy of bystanders, we automatically detected and blurred all faces using deface [105]. Examples of blurred images are shown in Figure 6.

In conclusion, we have addressed data-related concerns by compensating participants, obtaining signed consent for data recording, preserving privacy through face blurring, and ensuring no personal information is disclosed. We believe our work will not result in any harmful consequences or negative societal impact.

# G   License, Data Accessibility and Maintenance

The data including its documentation will be released under the CC BY-NC-SA license and is available at `https://siplab.org/projects/EgoSim`. The dataset is composed of PNG images and CSV files, which are in open and widely used formats, ensuring ease of access and usability. Ground truth joint poses of the synthetic data in the SMPL-X format can be obtained via the AMASS website. Detailed explanations on how to read and utilize the dataset are provided on the hosted website. Upon acceptance, the code for EgoSim and our method will be released on GitHub under the GPL-3.0 license. The dataset and code will be hosted on ETH servers, ensuring long-term

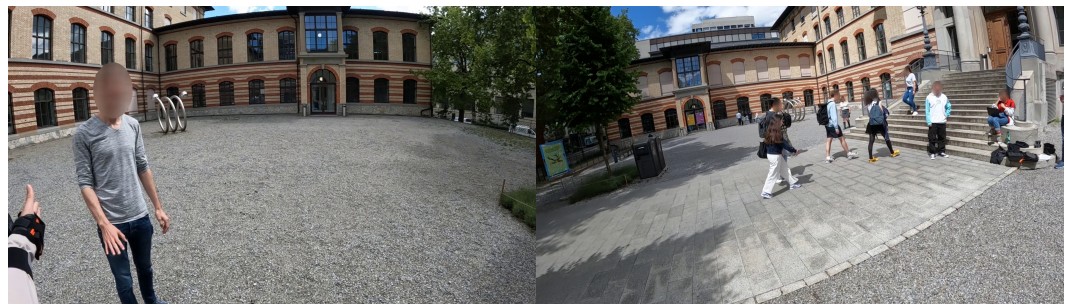

Figure 6: MultiEgoView typically does not feature many people in the field of view. We preserve people's privacy by automatically blurring their faces.

preservation and availability. The authors confirm that the data was collected consensually and bear all responsibility for any rights violations related to the dataset.

