# OpenReview forum: "EgoSim: An Egocentric Multi-view Simulator and Real Dataset for Body-worn Cameras during Motion and Activity"
_NeurIPS.cc/2024/Datasets_and_Benchmarks_Track — NeurIPS 2024 Track Datasets and Benchmarks Poster_

### Official Review · Reviewer_ppfE · 2024-07-16
**Interesting datasets that simulate body-worn cameras to understand the scene and user better from different perspectives to supplement real data captured in the same setting**

**Rating:** 7
**Confidence:** 4
**Correctness:** The claims are correct.

**Review:**

I believe this is the first paper (at least from what I have read) that proposes the use of multiple body-worn cameras to supplement current egocentric datasets which are mainly captured from head-mounted cameras. On top of different perspectives, the authors also provide different modalities that are common for synthetic datasets. These can help to improve egocentric perception research, especially for egocentric pose estimation task as well as related field such as hand-object interaction, but probably not addressed in this paper. The authors have made efforts to ensure the simulations are as realistic as possible with the use of captured 3D poses and physics simulation of cameras movement during body motions. The simulator has been used to generate a large synthetic dataset and a real-world dataset is also collected. A 3D pose estimation experiment has been conducted to test the performance of using the multiple perspectives and synthetic/real datasets. From the experiment, the authors demonstrated that a mixture of synthetic and real data provides the best results. The limitations of the paper are discussed in the paper such as the type of animations that can be extracted from AMASS and the limited virtual scenes that the simulator can currently support. There are also suggestions from the authors to consider other methods to improve pose estimations and expand the datasets for other perception tasks such as depth estimation, semantic segmentation, which I agree with. I would like to add that it will also be interesting to explore the cross-view translation so that existing egocentric datasets can use the simulator to generate views from other parts of the body.

**Strengths:**

This research is very useful for egocentric perception and is novel in the introduction of simulating body-worn cameras. While it may not be realistic to an end user to don cameras on four limbs and pelvic for daily tasks, I can appreciate the use of multiple views and modalities to achieve better egocentric 3D pose estimation. The authors made good design considerations for the simulator to ensure the synthetic data is as realistic as possible. The experiment conducted is sufficient to prove the use of both synthetic and real data for better pose estimation. It will be better if the authors also conducted an experiment to compare the results from single perspective as this will help us to understand the effect of multiple perspectives.

**Additional Feedback:**

I look forward to the future research that will span from this dataset.

**Clarity:**

The paper is clear. Table 2 can be improved by bolding the best results and also provide arrows next to the metrics used so that readers can easily know where they are higher=better or lower=better.

**Documentation:**

Yes. Details are provided in main paper, supplementary materials and website.

**Ethics:**

No ethical concerns. Human data collected does not cause harm to the subjects and data is anonymized.

**Limitations:**

Authors have adequately addressed the limitations and I have given my suggestions above.

**Opportunities For Improvement:**

More experiments can be conducted to compare single vs multiple perspectives as an example. It is expected that performance will improve but it is not know by how much.
Human motions are limited by the use of AMASS and the authors addressed this in the paper. A suggestion will be to consider generative motion models such as Motion-GPT and human/hand object interaction datasets.
The virtual scenes supported can be expanded to include datasets that has customizable virtual environments such as Habitat or BEHAVIOR.

**Relation To Prior Work:**

Authors addressed this well enough in the "Related Work" section and provided Table 1 for comparison with existing datasets.

**Summary And Contributions:**

The authors developed EgoSim which simulates images captured from body-worn cameras placed on the four limbs and pelvis, in addition to head-mounted cameras (which is the most common perspective for egocentric datasets). Real motion sequences captured from AMASS dataset is used to ensure that the motions are realistic and physics simulation is used to estimate the forces on the cameras during motion to produce motion artifacts. With the simulator, a synthetic dataset has been generated (77-hour videos) and a 5-hour real-world dataset has been collected to test the use of multiple body-worn cameras images to perform egocentric 3D pose estimation and demonstrate the use of the synthetic data to complement real-world data. Analysis is also performed on the sim2real gap.

---

> ### Author Rebuttal · Authors · 2024-08-17
>
> Thank you for your review and for sharing your enthusiasm about future research in egocentric vision with body-mounted cameras.
>
> We believe that our ablation study on various subsets of camera configurations addresses your question about the benefits of multiple perspectives (see Table 4 in the supplementary material). The study highlights the added value of using multiple cameras and also includes results for a new single-camera configuration (see accompanying PDF) to provide a comprehensive comparison.
>
> We agree that the suggestion of integrating motion-generating models, such as MotionGPT, into EgoSim is promising and directly enabled by EgoSim’s support for any skeletal meshes in FBX format or SMPL. This flexibility not only future-proofs EgoSim by allowing it to adapt to emerging methods but also opens up opportunities for generating specific new tasks and benchmarks.
>
> Our additional 22 scenes address variability in the synthetic dataset (please see the attached PDF for example images). Future work will be fully able to utilize the API and assets from Unreal Market Place to effortlessly customize and randomize the virtual scenes, e.g. spawn and move objects.

---

> > ### Comment · Reviewer_ppfE · 2024-08-26
> >
> > The introduction of the 22 scenes will help to improve the synthetic dataset. Thank you for addressing my concerns.

---

### Official Review · Reviewer_Lax6 · 2024-07-22
**A simulation platform that can synthesize egocentric videos from body-worn cameras, and a synthetic dataset created by this platform, accompanied with real-world data collected in a similar setting**

**Rating:** 7
**Confidence:** 4
**Correctness:** Yes
**Clarity:** Yes

**Review:**

This paper studies the application of body-worn cameras to egocentric tasks like body pose estimation. As the authors pointed out, this is a promising direction considering the increasing miniaturization of optical sensors. Comparing with head-mounted cameras, body-worn cameras show a new perspective that can be helpful for better understanding the environment, and estimating the body pose, especially the lower body. This paper proposes EgoSim simulator that can synthesize realistic images from body-worn cameras by taking motion artifacts into consideration. A dataset synthesized by the proposed simulator, together with real-world data capture, can be successfully used to train a body pose estimator. The experiments can potentially be improved to better showcase the advantage of EgoSim.

**Strengths:**

1. This paper is well motivated. Head-mounted cameras are limited by the occlusion, and may not be ideal for many egocentric tasks, e.g., full body pose estimation. With the increasing miniaturization of optical sensors, using multiple body-worn cameras will be a feasible and promising solution. This paper is a valuable exploration into this direction.
2. The simulator, as well as the data generation and collection pipeline are clearly explained, making it reproducible.
3. This paper proposes an effective end-to-end body pose estimation method as a baseline. Experiments showcase the usefulness of the dataset on the egocentric body pose estimation task.

**Additional Feedback:**

N/A

**Documentation:**

Yes

**Ethics:**

No. The real-world data collection was reviewed and approved by the ETH Zurich Ethics Commission.

**Limitations:**

Yes. Limitations and potential negative impact are well discussed.

**Opportunities For Improvement:**

1. The most important feature of EgoSim should be the camera motion artifacts simulation, because rendering and camera synchronization are trivial in a simulation environment. However, the analysis or experiment on this feature is missing. Although a mass-spring system sounds reasonable for the motion artifacts simulation, it would be more helpful if we can have a better understanding of how it differs from real-world camera motion. Alternatively, it would be good to see an ablation study showing that the motion artifacts simulation is really helpful for improving the performance on the body pose estimation task.
2. As the authors mentioned, a major advantage of using simulation is to synthesize data in diverse environments. However, the MultiEgoView dataset consists of only two scenes, one of which is the 3D scan of the environment where the real-world capture happened. It is doubtful whether this setting can demonstrate the generalization ability of the body pose estimator. Since acquiring 3D scans for real-world environment is expensive and unscalable, the suggested usage of EgoSim should be generating data in massive diverse environments. It would be good to see an experiment that the model is trained solely using synthetic data, and can generalize well to real-world tasks.

**Relation To Prior Work:**

Yes

**Summary And Contributions:**

This paper proposes a simulator that generates realistic renderings from multiple body-worn cameras given motion capture data. Considering the loose attachment of body-worn cameras, the simulator can generate motion artifacts to mimic the slip and drag of cameras’ positions. This paper also proposes a dataset consisting of synthetic data generated by the proposed simulator, as well as real-world data collected in a similar setting. The dataset is empirically shown to be useful on training a body pose estimator.

---

> ### Author Rebuttal · Authors · 2024-08-17
>
> We thank you for the review of our paper and will address your questions and suggestions.
>
> We agree that a more diverse synthetic dataset is more suited to ensure robustness and sim-2-real performance. To this end, we have rendered an extra 33h of 2 scenes and 22 unique locations with considerable variety: tennis courts, city, parking lot, park, … (see the unreal assets [1,2] and the attached PDF).
>
> We are currently still training models on the large number of new frames. Once complete, we will provide an ablation study to showcase the effect of the extra data on sim-2-real/cross-scene accuracy and report the updated results.
>
> Our combination of extensive synthetic and manually collected real data provides a solid foundation for multi-camera egocentric pose estimation. For novel tasks or specific needs, researchers may leverage EgoSim to synthesize additional data, benefiting from the extensive Unreal Engine asset marketplace that is fully compatible with EgoSim. Additionally, our real-world dataset can serve as a valuable resource for verifying and enhancing the effectiveness of domain adaptation techniques.
>
> Finally, we agree that an ablation study on the effect of the motion artifacts and spring system would be insightful. We will provide this analysis once training and validation are completed on the new scenes.
>
> [1] https://www.unrealengine.com/marketplace/en-US/product/city-park-environment-collection
>
> [2] https://www.unrealengine.com/marketplace/en-US/product/city-sample-buildings

---

> > ### Author Rebuttal · Authors · 2024-08-28
> >
> > We appreciate your patience as we prepared our response regarding the appropriateness of using a spring-damper system to model real camera motion.
> >
> > To address this, we conducted an ablation study to examine the similarity between real camera motions and those simulated using a spring-damper system. We recorded the wrist movements and the attached camera using a motion tracking system (https://optitrack.com/cameras/primex-13/). We then compared the motion path error between the real camera motion and two scenarios: 1) a synthetic camera track with a rigidly mounted camera and 2) a camera attached via a spring-damper system.
> >
> > As illustrated in our results table and the qualitative analysis provided in the supplementary PDF, the camera trajectory generated using the spring-damper system demonstrates a closer alignment with real camera motion. We compute the similarity between the trajectories with the DTW-aligned mean Euclidean distance. The spring system yields a 16\% lower mean alignment error.
> >
> > Notably, in scenarios involving strong outward acceleration and rapid directional changes, the spring-damper system more accurately replicates the inertia-induced slight movements observed in real camera motion, resulting in a more realistic simulation.
> >
> > We will include the plots and quantitative results from the attached PDF in our revision.

---

### Official Review · Reviewer_C1rr · 2024-07-22
**EgoSim is a novel simulator for body-worn cameras, generating realistic egocentric renderings from multiple perspectives using real motion capture data.**

**Rating:** 7
**Confidence:** 4
**Correctness:** Yes
**Clarity:** Yes

**Review:**

Quality

The paper introduces the EgoSim simulation platform, which generates realistic egocentric renderings from multiple camera perspectives using motion capture data and physical simulation.
The MultiEgoView dataset includes 77 hours of simulated footage and 5 hours of real-world data, providing a comprehensive testbed for training and evaluating models.

Clarity

The paper is well-structured with clear sections explaining the simulator, dataset, and experimental results.
The authors provide thorough technical details, making the paper accessible to readers with varying levels of expertise.
Originality
The use of body-worn cameras at multiple locations is a novel approach compared to traditional head-mounted setups.
The simulation of motion artifacts due to non-rigid camera attachments is a unique feature that enhances the realism of the generated data.

Significance

This work significantly contributes to the field of egocentric perception, particularly in human motion tracking and pose estimation.
The findings have practical implications for the development of AI-based personal devices and wearable technology.
Pros

The combination of synthetic and real-world data in the MultiEgoView dataset enhances its robustness.
The inclusion of motion artifacts and diverse environments in the simulations improves their realism and applicability.
The detailed methodology makes it easier for other researchers to replicate and build upon the work.

Weakness

The current version of EgoSim does not account for interactions with other humans or objects, which could limit its applicability in certain scenarios.
The simulator currently features only two scenes, which may not fully capture the diversity of real-world environments.
The model shows higher error in global root position estimates, indicating room for improvement in this area.

**Strengths:**

The EgoSim simulator provides a highly realistic and configurable platform for generating egocentric multi-view footage, which is valuable for training and evaluating computer vision models for tasks like body pose estimation.

The MultiEgoView dataset is a contribution, providing both simulated and real-world egocentric data with corresponding 3D body poses, which can enable significant advances in this research area.

The authors demonstrate the benefits of their simulator and dataset by training a video-only pose estimation model, showing the domain gap between simulated and real-world data, and the value of their contributions in bridging this gap.

**Additional Feedback:**

Here are some additional suggestions for improving the paper:

Provide experiments with existing methods on the dataset:

Implement and evaluate the performance of relevant existing methods or baselines on the proposed dataset. This will help readers better understand the characteristics of the dataset and establish a performance benchmark.
Analyze the strengths and limitations of the existing methods when applied to this dataset, and discuss how the proposed approach aims to address the identified shortcomings.

Elaborate on the use of six cameras:

Explain in more detail why six cameras were chosen for data collection, and how this design decision relates to the intended use cases or challenges being addressed.
Discuss any tradeoffs or design considerations that went into selecting the number and placement of cameras.
Describe how the multi-camera setup contributes to the dataset's unique properties or capabilities compared to single-camera datasets.

Provide more details on the EgoSim Simulation Platform:

Give a comprehensive overview of the EgoSim platform, including its architecture, key features, and capabilities.
Explain how the simulation platform was used to generate or augment the dataset, and describe the benefits or advantages it provides compared to other simulation tools.

**Documentation:**

The paper clearly states that EgoSim is an open-source platform, indicating its availability and accessibility to the research community.
The authors mention that EgoSim is accompanied by comprehensive documentation, tutorials, and sample scenarios to assist users in utilizing the platform.

**Limitations:**

The authors should discuss the motivation of the six-camera setup further.

**Opportunities For Improvement:**

Please provide experimental results and analysis on how existing state-of-the-art methods perform on the proposed dataset.

How realistic is the six-Camera setup? Why should we use the six-camera setup?

Please clearly explain the benefits and advantages of using a six-camera setup for this dataset. Why should we need to attach camera on foot?

Discuss how the multi-view information and increased coverage provided by the six cameras can contribute to addressing specific research problems or enabling new applications.

Please quantify the performance improvements or insights gained by leveraging the six-camera setup compared to a single-camera or fewer-camera configuration.

**Relation To Prior Work:**

Yes

**Summary And Contributions:**

Summary

This paper introduces EgoSim, a novel simulator for body-worn cameras that generates realistic egocentric renderings from multiple perspectives across a wearer’s body. It also presents MultiEgoView, a dataset of egocentric footage from six body-worn cameras and 3D body poses during various activities

Contributions

1. EgoSim Simulator: An adaptable and realistic simulator for multiple body-worn cameras using real human motion data.
2. MultiEgoView Dataset: A comprehensive dataset with 77 hours of simulated footage and 5 hours of real-world data from 13 participants.
3. Pose Estimation Method: A learning-based multi-view method for end-to-end training on video for 3D pose estimation tasks.
4. Sim-to-Real Transfer: Demonstrates the effectiveness of using simulated data to improve real-world pose estimation accuracy.

---

> ### Author Rebuttal · Authors · 2024-08-17
>
> Thank you for your review. We will clarify your questions about the benefits of the 6-body-worn camera setup.
>
> The 6-camera configuration (head, pelvis, and limbs) is aligned with established practices in pose estimation research from body-worn sensors. Specifically, significant research on inertial sensor-based pose estimation commonly uses the 6-sensor configuration we mirrored (head, pelvis, limbs) and has shown its effectiveness [1,2,3,4].
>
> On top of the established standard from the inertial pose estimation community, each mounting position of the camera offers specific advantages: the head aligns with the gaze direction, the pelvis provides stable environmental observation, the wrists capture close-up hand interactions, and the legs monitor stride and ground contact and could thus prevent foot skating effects. We will reiterate these advantages in our revision.
>
> We have showcased the benefits of the 6-camera setup over a head+wrists and limbs in an ablation study in the supplementary material and added a new head-only configuration (global MPFJPE 0.18/0.238/0.293/0.345, see attached PDF).
> We will also add this evaluation to our revised paper.
>
> Because no previous benchmarks on body-pose estimation from multi-view body-worn cameras existed, we could not directly transfer baseline algorithms for this specific topic for comparison in our submission.
>
> About the overview of the EgoSim platform:
>
> Section 3 of our paper and Table 1 in the supplementary material provide an overview of our platform. We will add further details in our revised paperto highlight the differentiating factors from other simulators. If you prefer additional information on another aspect, we appreciate any feedback.
>
> We again thank you for your comments and hope we have addressed them to your satisfaction.
>
>
> [1] Yi, Xinyu, et al. "Physical inertial poser (pip): Physics-aware real-time human motion tracking from sparse inertial sensors." Proceedings of the IEEE/CVF conference on computer vision and pattern recognition. 2022.
>
> [2] Ponton, Jose Luis, et al. "SparsePoser: Real-time full-body motion reconstruction from sparse data." ACM Transactions on Graphics 43.1 (2023): 1-14.
>
> [3] Yi, Xinyu, Yuxiao Zhou, and Feng Xu. "Transpose: Real-time 3d human translation and pose estimation with six inertial sensors." ACM Transactions On Graphics (TOG) 40.4 (2021): 1-13.
>
> [4] Huang, Yinghao, et al. "Deep inertial poser: Learning to reconstruct human pose from sparse inertial measurements in real time." ACM Transactions on Graphics (TOG) 37.6 (2018): 1-15.

---

> > ### Comment · Reviewer_C1rr · 2024-08-24
> >
> > Dear authors,
> >
> > Thanks for addressing my concerns. I have increased the rating from 6 to 7. Good luck with your work.
> >
> > Best regards,
> >
> > Reviewer

---

### Official Review · Reviewer_rQh4 · 2024-07-24

**Rating:** 6
**Confidence:** 4
**Correctness:** See "Limitations" and "Strengths"
**Clarity:** See "Summary" and "Strenghts"

**Review:**

See "Summary" and "Limitations"

**Strengths:**

- The paper introduces EgoSim, a novel simulator that extends existing tools by focusing on body-worn cameras, offering new perspectives for egocentric tasks in computer vision.
- MultiEgoView provides a significant amount of data, both synthetic and real, which can be valuable for various research and development applications in pose estimation and human motion analysis.
- The use of real motion capture data and physical simulation of camera attachments enhances the realism of the generated footage, making it more useful for training models that can generalize well to real-world data.
- The paper provides a thorough analysis of the sim-to-real gap and demonstrates the benefits of using synthetic data to improve real-world pose estimation performance.

**Additional Feedback:**

- How do the authors plan to expand the real-world data collection to include more diverse environments and activities?
- Can you provide more details about the diversity of the participants involved in the real-world data collection? How do their demographics and activities ensure the robustness of the dataset?
- What steps are being taken to increase the variety of environments in which the real-world data is collected? Would expanding this aspect help in better generalization of the models?
- Can you provide more details on the methodologies used to quantify the sim-to-real gap? What specific improvements are planned to reduce this gap further?
- How was the synthetic data validated against real-world scenarios to ensure its realism? Can you provide more comparative analyses?

# Minor comments:
-  In line 154 there is a typo, it should be 49.8 M

**Documentation:**

Yes.

**Ethics:**

No.

**Limitations:**

- While the dataset includes 5 hours of real-world data, this is relatively small compared to the 77 hours of synthetic data. The validation of realism could be enhanced by comparing the generated data with more extensive real-world data.
- While the real-world data collection includes 13 participants, the demographic diversity in terms of ethnicity and broader age groups is limited. This might affect the generalizability of the findings.
- The paper primarily addresses pose estimation. While this is a critical task, broader applications of the dataset and simulator could be explored.

**Opportunities For Improvement:**

See "Limitations"

**Relation To Prior Work:**

Yes.

**Summary And Contributions:**

The paper "EgoSim: An Egocentric Multi-view Simulator for Body-worn Cameras during Human Motion" introduces EgoSim, a simulator designed to generate realistic egocentric renderings from multiple perspectives across a wearer's body using real motion capture data and a physical simulation of camera attachments. The authors also present MultiEgoView, a dataset containing 77 hours of simulated footage and 5 hours of real-world motion data from 13 participants, to train and benchmark video-only pose estimation models. The paper demonstrates that combining synthetic and real data can enhance the training of these models.

---

> ### Author Rebuttal · Authors · 2024-08-17
>
> We appreciate the thorough review!
> We acknowledge your point on dataset diversity for robustness. AMASS encompasses a broad range of human body shapes across different age groups, which EgoSim leverages to generate renders based on these diverse motions and activities. Similarly, our real-world dataset includes participants with a wide range of heights (160–190 cm tall) and body weights (50–94 kg heavy). Since the cameras are outward-facing, our recordings do not capture any bias or effect based on the ethnicity of our participants.
> For our synthetic data, where the cameras capture other characters within their field of view, we ensure a fair representation across ethnicities by rendering avatars with randomly sampled skin colors and clothing.
> For recording our real-world dataset, participants performed motions that correspond to the most commonly found motions according to the BABEL motion annotation [1]. We expect them to be relevant to the community as diverse and representative of a wide range of different motion types.
>
> How we plan to increase the variety of environments for real-world data:
>
> We acknowledge this question, which was the primary motivation for developing EgoSim: the time-consuming nature of multi-camera recordings.  Given the flexibility and scalability of our simulator, the purpose of our real-world dataset MultiEgoView, the first 6-camera dataset for pose estimation, was to verify EgoSim’s ability to minimize the need for extensive real-world data collection and to identify potential gaps. We demonstrate that the combination of synthetic and real-world data is beneficial and reduces the need for expensive real-world data collection (see Table 2). Our real-world dataset comprises recordings of ~5h @ 30Hz (6 x 540k = 3.2M frames) from 13 participants. This is comparable to other “first-of-a-kind” datasets for pose estimation, e.g., TotalCapture with 5 participants [2], Mo2Cap2 [3] with 2.7k frames and [4] with 27k frames for egocentric pose estimation with a single head-mounted fisheye camera. As we are the first to address the multiple body-mounted camera pose estimation, we provide our model as the first “baseline” in that direction.
>
> We acknowledge a remaining domain gap between the synthetic and real data, which can lead to reduced accuracy when relying solely on synthetic training data. However, we also show that even with little real-world training data that gap gets significantly smaller (see Table 2). We will further discuss this limitation but also the benefit of already little real-world data in our revision.
>
> We aim to report the updated results for the sim-2-real performance with our extended synthetic data set of 22 additional scenes (as detailed above). The model is currently still training.
>
> Overall, we are confident that our real-world dataset captures a diverse and representative range of human motions, and our participants reflect a broad demographic. The integration of a large, varied synthetic dataset, which includes multiple locations, body shapes, appearances, and motions, further strengthens the dataset's robustness. The addition of 22 new scenes enhances its capability to handle a wide variety of surroundings, ensuring that our dataset is both comprehensive and adaptable for future research and applications.
>
> [1] Punnakkal, Abhinanda R., et al. "BABEL: Bodies, action and behavior with english labels." Proceedings of the IEEE/CVF Conference on Computer Vision and Pattern Recognition. 2021.
>
> [2] Trumble, Matthew, et al. "Total capture: 3D human pose estimation fusing video and inertial sensors." BMVC. Vol. 2. No. 5. 2017.
>
> [3] Xu, Weipeng, et al. "Mo 2 cap 2: Real-time mobile 3d motion capture with a cap-mounted fisheye camera." IEEE transactions on visualization and computer graphics 25.5 (2019): 2093-2101.
>
> [4] Wang, Jian, et al. "Scene-aware egocentric 3d human pose estimation." Proceedings of the IEEE/CVF Conference on Computer Vision and Pattern Recognition. 2023.

---

> > ### Comment · Reviewer_rQh4 · 2024-08-26
> > **Keeping my scores**
> >
> > Thank you for your rebuttal, I am keeping my scores.

---

### Author Rebuttal · Authors · 2024-08-17

We thank all reviewers for their time and valuable feedback. We were glad to read that reviewers appreciate the “valuable exploration” (**Lax6**) into estimating pose from multiple body-worn cameras, for which we provide a “significant amount of data, both synthetic and real” (**rQh4**). We are happy to hear that we could show that “EgoSim [...] provides a highly realistic and configurable platform for generating egocentric multi-view footage” (**C1rr**) and that “[t]he experiment conducted is sufficient to prove the use of both synthetic and real data for better pose estimation.” (**ppfE**).

We now address reviewers’ concerns. We also respond to the reviews individually for further discussion.

The main contribution of our paper is the first ego-centric dataset from 6 body-worn including real and synthetic data. This dataset is accompanied by a fully configurable simulator for a customizable number of virtual cameras, freely positionable on a virtual avatar’s body with customizable intrinsics and multi-modal output, where avatar motions originate from human motion capture recordings to ensure realistic movements. With our simulator, we rendered a virtual 6-camera dataset across (now) 4 large scenes for training and testing a baseline model, so that future efforts may choose to directly use our simulated rendered data, re-render their own using our simulator code release, or use our real-world dataset.

Below, we aim to clarify questions about our dataset, comprising synthetic as well as real data, and the capabilities and use of the EgoSim platform.

## DIVERSITY OF SYNTHETIC DATA AND SIM-2-REAL GAP (**C1rr**, **Lax6**)

EgoSim supports effortless integration of new scenes. In response to the reviewers' comments on the diversity of the synthetic data, we have expanded our dataset by adding 21 million additional frames captured across 2 scenes in 22 unique locations. This expansion recognizes the importance of Domain Randomization and data diversity for enhancing sim-to-real performance and robustness [1].
The 33h of new scenes encompass a wide variety of environments, including urban streets, tennis, basketball, baseball, and football courts, as well as parks both with and without vegetation, and lakes (refer to the Unreal assets [1,2]). This shows the flexibility and versatility of our simulator. We will include images of all scenes in the supplementary material (please see a preview of the new scenes in the attached PDF).



## BENEFIT OF MULTI-VIEW BODY-WORN CAMERAS (**C1rr**, **Lax6**, **ppfE**)

Reviewers **C1rr**, **Lax6**, and **ppfE** inquired about the benefits of the six-camera layout. To address their questions, we conducted an ablation study that demonstrates that the six-camera configuration significantly outperforms setups that use only the head and wrist cameras or cameras mounted on all limbs. The detailed results of this study can be found in Table 4 of the supplementary material. Additionally, we now include the results for the head-only configuration (please see the attached PDF document). This configuration yielded a global MPJPE of 0.345, compared to 0.18 when using all six cameras.

We hope that these comments and new results provide additional clarity on the benefits of our approach and address the reviewers' concerns. We will be happy to respond to further comments during this discussion phase.

[1] https://www.unrealengine.com/marketplace/en-US/product/city-park-environment-collection

[2] https://www.unrealengine.com/marketplace/en-US/product/city-sample-buildings

---

> ### Author Rebuttal · Authors · 2024-08-22
>
> We thank the reviewers for waiting for our updated results that report now.
>
> We conducted additional experiments by training a model using all 24 scenes, including 22 newly introduced scenes alongside the original 2. This approach resulted in a significant 0.25 reduction in global MPJPE, effectively narrowing the sim-to-real gap compared to the model trained on just 2 scenes. These findings underscore the importance of increasing the diversity within the synthetic dataset to enhance model generalization in real-world scenarios. Additionally, the expanded training set led to a 0.03 improvement in global MPJPE on synthetic data, highlighting the enhanced performance even within the synthetic domain. Furthermore, by fine-tuning with 20% of our real dataset, the global MPJPE on real data was further reduced from 0.44 to 0.4, demonstrating the benefits of using a more diverse set of scenes.
>
> For detailed results, please refer to the table below, which will be included in the final revision. We hope this addresses any questions regarding the diversity of the synthetic data. We remain open to further inquiries and are happy to provide additional clarification if needed.
>
>
> | Trained on                          | evaluated on                   | Global MPJPE ↓ | PA-MPJPE ↓ | MTE ↓ | MRE ↓ | MJAE ↓ | Jerk |
> |-------------------------------------|--------------------------------|---------------|------------|-------|-------|--------|------|
> | Synthetic Synthetic (2 scenes)      | Synthetic (2 scenes)           | 0.18          | 0.041      | 0.14  | 0.334 | 9.3    | 21.7 |
> | **NEW** Synthetic (24 scenes)       | Synthetic (2 scenes)           | 0.15          | 0.039      | 0.12  | 0.271 | 9.0    | 22.2 |
> | Synthetic Synthetic (2 scenes)      | Real                           | 1.02          | 0.203      | 0.98  | 1.081 | 33.9   | 15.8 |
> | **NEW** Synthetic (24 scenes)       | Real                           | 0.77          | 0.119      | 0.71  | 0.947 | 29.0   | 20.9 |
> | Real                                | Real                           | 1.23          | 0.087      | 0.79  | 1.030 | 16.4   | 1.5  |
> | *with fine-tuning:*                 |                                |               |            |       |       |        |      |
> | Synthetic + 20% Real                | Real                           | 0.44          | 0.060      | 0.41  | 0.562 | 13.1   | 15.9 |
> | **NEW** Synthetic 24 + 20% Real     | Real                           | 0.40          | 0.056      | 0.37  | 0.504 | 12.8   | 15.4 |
> | Synthetic + 80% real                | Real                           | 0.36          | 0.047      | 0.34  | 0.486 | 10.5   | 16.7 |
> | Synthetic + 10 real participants    | Real (cross-participant)       | 0.52          | 0.068      | 0.49  | 0.651 | 17.2   | 13.5 |

---

### Decision · Program_Chairs · 2024-09-26

**Decision:**

Accept (Poster)

**Comment:**

All reviewers agreed to accept the paper. The AC also supports accepting the paper and strongly suggests the authors to follow the reviewers’ feedback in the camera-ready version.